# Is Robustness Transferable across Languages in Multilingual Neural Machine Translation?

**Leiyu Pan, Supryadi and Deyi Xiong**[*]
College of Intelligence and Computing, Tianjin University, Tianjin, China
{lypan, supryadi, dyxiong}@tju.edu.cn

## Abstract

Robustness, the ability of models to maintain performance in the face of perturbations, is critical for developing reliable NLP systems. Recent studies have shown promising results in improving the robustness of models through adversarial training and data augmentation. However, in machine translation, most of these studies have focused on bilingual machine translation with a single translation direction. In this paper, we investigate the transferability of robustness across different languages in multilingual neural machine translation. We propose a robustness transfer analysis protocol and conduct a series of experiments. In particular, we use character-, word-, and multi-level noises to attack the specific translation direction of the multilingual neural machine translation model and evaluate the robustness of other translation directions. Our findings demonstrate that the robustness gained in one translation direction can indeed transfer to other translation directions. Additionally, we empirically find scenarios where robustness to character-level noise and word-level noise is more likely to transfer.

## 1 Introduction

The research interest in multilingual neural machine translation (MNMT) has been increasing in recent years (Zhang et al., 2020; Yigezu et al., 2021; Costa-jussà et al., 2022). This is partially due to the capability of MNMT in translating multiple language pairs, including both high- and low-resource languages, with a single model, easing the deployment of machine translation systems. Despite this capability, MNMT, like bilingual neural machine translation, is still vulnerable towards domain shift (Müller et al., 2020), perturbations and noises in words and sentences (Belinkov and Bisk, 2018). Furthermore, low-resource language

pairs in MNMT are more vulnerable to perturbations than high-resource language pairs, which is observed in our preliminary experiments.

In the context of NLP, transfer learning has proven highly effective in various tasks, including sentiment analysis (Liu et al., 2019), named entity recognition (Lee et al., 2018), and question answering (Chung et al., 2018). Existing research demonstrates that transfer learning is also present in MNMT models (Lakew et al., 2018), e.g., knowledge transfer from high-resource language translation to low-resource language translation. While previous research has shed light on the effectiveness of knowledge transfer within different language pairs, the robustness transfer across diverse translation directions remains an open question.

In this work, we aim to investigate whether and how robustness is transferable across languages in MNMT. To achieve this purpose, we attack the inputs for a particular translation direction and then observe the difference in performance of other translation directions on the noise test set. The model that we use is a standard multilingual translation model which shares the Transformer encoder-decoder between language pairs. The adversarial attack is conducted in both one-to-many and many-to-one translation tasks. Meanwhile, we select languages from the same and different language families for our experiments. For each language pair, we generate 3 types of noises (character-, word-, and multi-level noise) in the source language.

After attacking the MNMT model with different noises in the source language, we evaluate the translation quality of the language pair that is attacked as well as that of other language pairs that are not attacked in terms of BLEU. We are especially interested in the impact of the adversarial attack to an MNMT model on language pairs that are not attacked.

To summarize, the contributions in this paper are:

---

[*]Corresponding author.

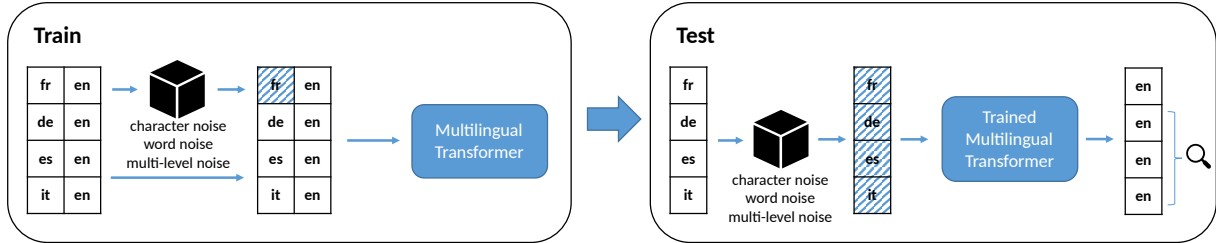

Figure 1: The proposed robustness transfer analysis protocol. Slashed cells indicate languages being attacked in a black-box way.

- We propose a robustness transfer analysis protocol to investigate the robustness transfer phenomenon in MNMT models.

- We have empirically found that the attack with character/word/multi-level noise increases the robustness of the MNMT model.

- We have further observed that robustness can transfer across languages. The robustness of the character-level noise is more likely to transfer across related languages while the robustness of word-level noise is more likely to transfer across distant languages.

## 2 Related Work

MNMT supports one-to-many, many-to-one, and many-to-many translation via partial or full parameter sharing (Ha et al., 2016; Firat et al., 2016; Johnson et al., 2017). Massively MNMT has been explored, where a single model supports a large number of languages, including high- and low-resource languages (Aharoni et al., 2019). In the context of massively MNMT, M2M-100, a non-english-centric MNMT model is proposed (Fan et al., 2021), which is capable of translating 100 language pairs. The successor to M2M-100, NLLB-200, hugely extends the number of languages to 200 via model scaling (NLLB Team et al., 2022).

NMT is vulnerable towards both naturally occurring and synthetic noise, e.g., human typos, character swaps (Belinkov and Bisk, 2018). Many works have been done to improve the robustness of NMT, including black-box approaches (Belinkov and Bisk, 2018; Niu et al., 2020; Zhang et al., 2021) and white-box methods (Cheng et al., 2018). Black-box attack methods include both character- and word-level attacks (Karpukhin et al., 2019). Character-level attacks are obtained usually by random character insertion, deletion, replacement and swapping, and word-level attacks can be performed

via similar word insertion and replacement (Alzantot et al., 2018) and so on. Differently, white-box attacks are usually gradient-based (Cheng et al., 2019).

Recent research has attempted to address the challenging task of enhancing the robustness of MNMT models. Zhou et al. (2021) propose robust optimization methods to tackle the issue of imbalanced data distribution between high- and low-resource languages. Grosso et al. (2022) attempt to address variations in data distributions across different domains by introducing robust domain adaptation techniques. Different from the above works, our work focuses on the robustness of MNMT models towards noise and the transferability of robustness across languages in MNMT models.

## 3 Robustness Transfer Analysis Protocol

In order to explore the phenomenon of robustness transfer across translation directions, we propose a robustness transfer analysis protocol. As shown in Figure 1, the protocol consists of 2 phases, a training phase and a testing phase.

During the training phase, we conduct black-box attacks on the source side within a specific translation direction in a multilingual training corpus. This entails carrying out character-level, word-level, or multi-level black-box attacks. Specifically, character-level black-box attacks involve random character insertion, deletion, replacement, and adjacent character swap operations. Word-level black-box attacks consist of random word swapping, deletion, insertion and replacement operations. For word replacement, we use pre-trained word embeddings to find words semantically related to the target word for replacement. The combination of character-level and word-level black-box methods constitutes the multi-level black-box approach. It is important to note that while we attack the source data in a particular translation direction, the target

data in the same translation direction remains unchanged, as well as the data in other translation directions. Lastly, the combined multilingual data are fed into the multilingual Transformer for training.

During the testing phase, we conduct a black-box attack on the source side of all translation directions on the multilingual test corpus. To maintain consistency in the distribution of the training and test data, we apply the same black-box attack operations to the test corpus as we do on the training corpus. The attacked source data are then fed into the multilingual Transformer, which is trained in the previous phase. We measure the translation performance of the translation directions that are not attacked during training. Additionally, we compare the translation performance of the model in this training setting with the translation performance of the model under other training settings. For instance, this can involve comparing it to the performance of a model trained on a clean corpus on the same test set, or to the performance of a model trained to perform other levels of black-box attacks on the same test set.

## 4 Experiments

With the robustness transfer analysis protocol, we conducted extensive experiments to investigate whether and how robustness transfers across languages in multilingual NMT.

### 4.1 Settings

**Data** We used the publicly available TED TALKS (Qi et al., 2018) and News Commentary as the datasets for training the multilingual translation model. To address the issue that Japanese-English parallel data in News Commentary are limited, we incorporated the KFTT dataset (Neubig, 2011) as additional resource. Our experiments covered both one-to-many and many-to-one settings with the dataset. In the many-to-one experimental setup, we specifically selected language pairs that include source languages from the same or different language families, allowing us to study the robustness transfer across different types of language pairs. We used the sentencepiece toolkit (Kudo and Richardson, 2018) to learn a shared vocabulary for all language pairs. The vocabulary size was set to 30,000.

**Attack** To implement character-level, word-level, and multi-level black-box attack, we utilized the nl-

paug library (Ma, 2019). For character-level black-box attacks, we set the proportion of attacked words to the total number of words in each sentence to 0.1. The coverage rate of the four character-level attack operations per sentence is 25%. Regarding word-level black-box attacks, the proportion of attacked words in each sentence was also set to 0.1 of all words. Words insertion and substitution operations were language-specific, where pretrained GloVe word embeddings were used for English[1] while pretrained fasttext word embeddings for other languages.[2] Specifically, we first extracted the top-k words with word embeddings similar to the target word, and then randomly chose one word from them as the insertion or replacement word. Similarly, the coverage of the four word-level attack operations per sentence is 25%. In the case of multi-level black-box attacks, character-level and word-level attack operations were combined. The coverage of eight attack operations per sentence was set to 12.5%.

**Model** We used the multilingual_transformer_i-wslt_de_en model provided by the fairseq framework (Ott et al., 2019) as our multilingual machine translation model. The hyperparameters we used to train the model are shown in Appendix B.

### 4.2 Is Robustness Transferable across Languages?

In the one-to-many and many-to-one experimental setups, we conducted separate experiments to study robustness transfer phenomenon. The results of the one-to-many experimental setup are shown in Table 1 while those of the many-to-one experimental setup are shown in Table 2 and Table 3.

**One-to-Many Translation**

In the one-to-many experimental setup, we introduced noise into the English-French translation direction while keeping the corpus of other translation directions unchanged. The results reveal that the performance of the non-attacked translation directions also improves on the test set with the added noise. For example, the model trained with character-level attacks on the English-French translation direction demonstrates better performance on the test set generated by character-level attacks compared to the model trained on a clean corpus (9.6 for clean corpus vs. 12.2 for character-level at-

---

[1]http://nlp.stanford.edu/data/glove.6B.zip
[2]https://fasttext.cc/docs/en/pretrained-vectors.html

| Training dataset | Test dataset | en-fr | en-ja | en-ar | en-de |
|---|---|---|---|---|---|
| clean corpus | clean corpus | **43.1** | 14.6 | **17.3** | **28.7** |
| | character-level attack | 28.0 | 9.6 | 9.3 | 16.6 |
| | word-level attack | 31.3 | 11.3 | 11.3 | 18.8 |
| | multi-level attack | 29.2 | 10.2 | 10.2 | 17.7 |
| character-level attack | clean corpus | 42.1 | 14.8 | 17.2 | 28.5 |
| | character-level attack | **38.8** | **12.2**(↑27.1%) | **13.1**(↑40.9%) | **22.2**(↑33.7%) |
| | word-level attack | 31.7 | 11.5 | 11.2 | 19.2 |
| | multi-level attack | 34.4 | 11.6 | 12.2 | 20.5 |
| word-level attack | clean corpus | 41.6 | 14.2 | 16.7 | 28.0 |
| | character-level attack | 30.5 | 10.2 | 9.5 | 17.3 |
| | word-level attack | **35.9** | 11.7(↑3.5%) | 12.3(↑8.8%) | 20.3(↑8.0%) |
| | multi-level attack | 32.8 | 10.5 | 10.5 | 18.4 |
| multi-level attack | clean corpus | 42.1 | **14.9** | 17.2 | 28.5 |
| | character-level attack | 37.4 | 12.0 | 12.8 | 21.8 |
| | word-level attack | 35.8 | **11.9** | **12.4** | **20.8** |
| | multi-level attack | **36.2** | **12.1**(↑18.6%) | 12.5(↑22.5%) | 21.1(↑19.2%) |

Table 1: BLEU scores for models with different training settings on different test sets using the TED TALKS dataset in the one-to-many case. The bolded scores represent the optimal performance on each test set for a specific language direction. The attacked translation direction is en-fr.

tack in English-Japanese translation direction). It is worth noting that models trained with black-box attacks also exhibit enhancements in performance on test sets that differ from the training data distribution. For example, models trained with word-level attacks on the English-French translation direction perform better on test sets generated by character-level attacks than models trained on a clean corpus. This can be attributed to the similarity between the character-level and word-level black box attack operations, as both involve insertions, replacements, deletions and swaps, thus resulting in mutual benefits. It is important to acknowledge that the one-to-many experimental setup implies that the source languages are all the same. Although we only attack the source language for a specific translation direction, the encoder might have learned a denoised representation of the source language during training. Thus, the difficulty of robustness transfer in the one-to-many experimental setup is relatively lower. Instead, it is more valuable to explore the phenomenon of robustness transfer across translation directions in a many-to-one scenario.

**Many-to-One Translation: Source Languages from the Same Language Family**

In the many-to-one experimental setup, we initially conducted robust transfer experiments where the source languages belong to the same language family, specifically the Indo-European language family. We incorporated noise into the French-English translation direction while keeping the corpus in other translation directions intact. The results of the experiments are presented in Table 2. Compared to the source languages being the same language, the source languages being different languages causes the robustness to be more difficult to transfer across translation directions. It can be observed that the percentage increase in BLEU scores in the not-being-attacked translation directions is not as high as that in the one-to-many experimental setting. However, it is undeniable that the not-being-attacked translation directions still show a significant performance improvement on the noisy test set. This is an even stronger evidence that robustness can transfer across translation directions. Similarly, we observe that the robustness of models trained with the character-level attack can also transfer to handle word-level noise in other translation directions, and vice versa.

For improving the reliability of our findings, we changed the attacked source language in the same target language setting. Specifically, we switched the attack direction from French-English to German-English and compared the performances across different attack direction cases. On the character-level noise test set, as depicted in Figure 2, we observe that the model's robustness to character-level noise can still transfer across trans-

| Training dataset | Test dataset | fr-en | de-en | es-en | it-en |
|---|---|---|---|---|---|
| clean corpus | clean corpus | **41.9** | 36.5 | 42.2 | **39.2** |
| | character-level attack | 24.2 | 23.8 | 27.7 | 25.8 |
| | word-level attack | 27.1 | 25.3 | 29.4 | 27.0 |
| | multi-level attack | 24.8 | 24.0 | 28.1 | 25.8 |
| character-level attack | clean corpus | 41.5 | **36.7** | **42.5** | **39.2** |
| | character-level attack | **37.9** | **26.2**(↑10.1%) | **31.2**(↑12.6%) | **29.1**(↑12.8%) |
| | word-level attack | 29.8 | 25.7 | 29.9 | 27.5 |
| | multi-level attack | 33.2 | 25.8 | 30.2 | 27.8 |
| word-level attack | clean corpus | 40.3 | 36.6 | 42.4 | 38.7 |
| | character-level attack | 28.1 | 24.4 | 28.2 | 26.0 |
| | word-level attack | **34.8** | 26.9(↑6.3%) | **32.3**(↑9.9%) | 29.2(↑8.1%) |
| | multi-level attack | 30.8 | 25.3 | 29.8 | 27.4 |
| multi-level attack | clean corpus | 41.0 | 36.5 | 42.1 | 38.9 |
| | character-level attack | 35.8 | 26.0 | 30.4 | 28.1 |
| | word-level attack | 34.8 | **27.0** | 32.1 | **29.4** |
| | multi-level attack | **35.1** | **26.2**(↑9.2%) | 30.8(↑9.6%) | 28.2(↑9.3%) |

Table 2: BLEU scores for models with different training settings on different test sets using the TED TALKS dataset in the many-to-one case. The bolded scores represent the optimal performance on each test set for a specific language direction. The source languages are from the same language family. The attacked translation direction is fr-en.

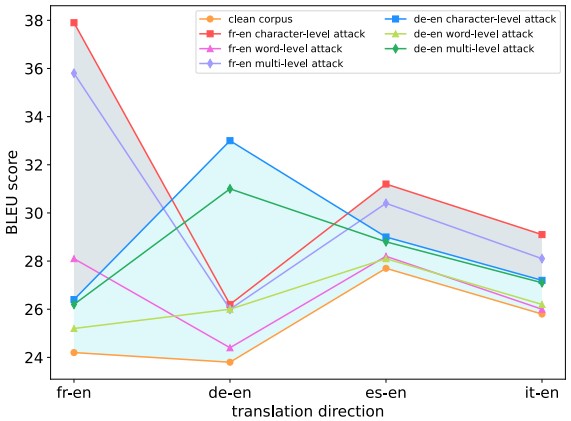

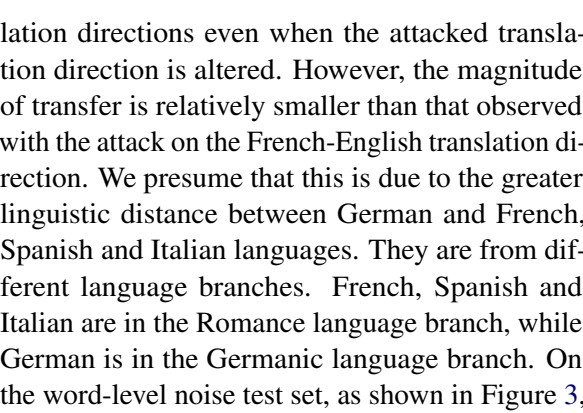

Figure 2: Comparison of the performance of attacking the fr-en translation direction and attacking the de-en translation direction on the character-level noise test set for various training settings.

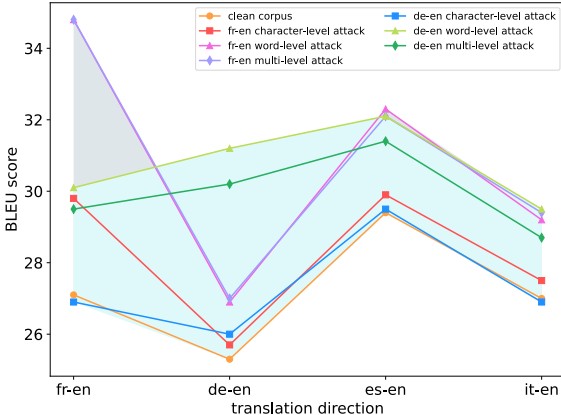

Figure 3: Comparison of the performance of attacking the fr-en translation direction and attacking the de-en translation direction on the word-level noise test set for various training settings.

lation directions even when the attacked translation direction is altered. However, the magnitude of transfer is relatively smaller than that observed with the attack on the French-English translation direction. We presume that this is due to the greater linguistic distance between German and French, Spanish and Italian languages. They are from different language branches. French, Spanish and Italian are in the Romance language branch, while German is in the Germanic language branch. On the word-level noise test set, as shown in Figure 3,

similarly we observe that word-level attacks on the German-English translation direction enhance the performance of other translation directions on the word-level noise test set. This provides additional evidence of the transferability of model robustness to word-level noise across translation directions. It is also noteworthy that the model robustness transfer degree for both word-level attacks are essentially the same. The BLEU scores of Spanish-English and Italian-English translation directions improve by roughly the same amount.

| Training dataset | Test dataset | fr-en | ja-en | ar-en | zh-en |
|---|---|---|---|---|---|
| clean corpus | clean corpus | **41.1** | 16.8 | 31.9 | 21.1 |
| | character-level attack | 22.5 | 14.8 | 22.8 | 20.3 |
| | word-level attack | 25.8 | 11.3 | 23.1 | 15.2 |
| | multi-level attack | 23.4 | 12.5 | 22.5 | 17.3 |
| character-level attack | clean corpus | 40.5 | 17.1 | 32.6 | **21.6** |
| | character-level attack | **37.1** | 15.6(↑5.4%) | 25.3(↑10.7%) | **21.0**(↑3.4%) |
| | word-level attack | 29.2 | 12.2 | 23.8 | 16.4 |
| | multi-level attack | 32.4 | 13.5 | 24.2 | 18.0 |
| word-level attack | clean corpus | 39.9 | **17.3** | 32.3 | 21.4 |
| | character-level attack | 28.1 | 15.5 | 24.3 | 20.7 |
| | word-level attack | **34.3** | **13.9**(↑23.0%) | 26.1(↑13.0%) | 17.8(↑17.1%) |
| | multi-level attack | 30.4 | 14.4 | 25.0 | 19.0 |
| multi-level attack | clean corpus | 40.3 | 17.1 | **32.8** | **21.6** |
| | character-level attack | 35.8 | **15.8** | **25.4** | 20.9 |
| | word-level attack | 34.0 | 13.5 | **26.3** | **17.9** |
| | multi-level attack | **34.5** | **14.4**(↑15.2%) | 25.7(↑14.2%) | **19.3**(↑11.6%) |

Table 3: BLEU scores for models with different training settings on different test sets using the TED TALKS dataset in the many-to-one case. The bolded scores represent the optimal performance on each test set for a specific language direction. The source languages are from different language families. The attacked translation direction is fr-en.

## Many-to-One Translation: Source Languages from Different Language Families

In the many-to-one experimental setup, we conducted robustness transfer experiments where the source languages are of different language families. In particular, French belongs to the Indo-European family, Japanese is in the Altaic family, Arabic comes from the Semitic family and Chinese is in the Sino-Tibetan family. We attacked the French-English translation direction, keeping the corpus in the other directions unchanged. Results are presented in Table 3. We can observe that the model's robustness to noise at all levels can also transfer across translation directions within this experimental setup. On the other hand, the robustness of the models to both character-level and word-level noise can also be transferred to each other and simultaneously across translation directions. It is noteworthy that the degree of robustness transfer across translation directions does not decrease significantly despite a decrease in source language similarity, which is a surprising finding.

## Many-to-One Translation: In-depth Analysis and Visualization

Furthermore, we delved into the reasons behind the transfer of robustness across translation directions. An example of an experimental setup with many-to-one and the source languages from the same language family was used for analysis. Firstly, we selected a seed sentence with the same meaning for each source language. Next, we generated four character-level noisy sentences for each source language's seed sentence using the character-level black-box attack method. These four sentences correspond to four character-level attack operations, including random character insertion, random character deletion, random character replacement, and random adjacent character swapping. Subsequently, we combined the four noisy sentences with the original seed sentences without noise from each source language. We chose two models for comparison: one trained on the clean corpus and the other trained on character-level attacks on the French-English translation direction. We then fed the prepared sentences into each of these models. Figure 4 visualizes dimension-reduced representations of the encoder outputs of these two models using PCA. We can observe that the representations of noisy sentences learned by the model trained on a clean corpus is more dispersed, whereas those learned by the model trained with character-level attacks on the French-English translation direction is more compact. This indicates that training with black-box attacks enhances the model's resilience to interference and reduces the susceptibility to biases caused by minor pertur-

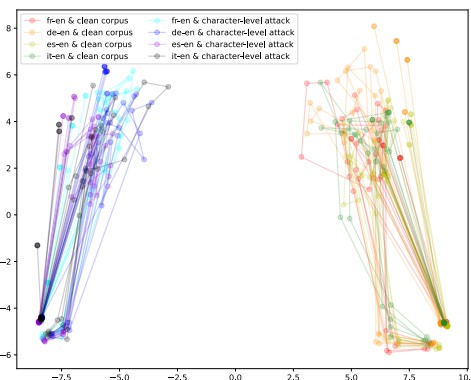

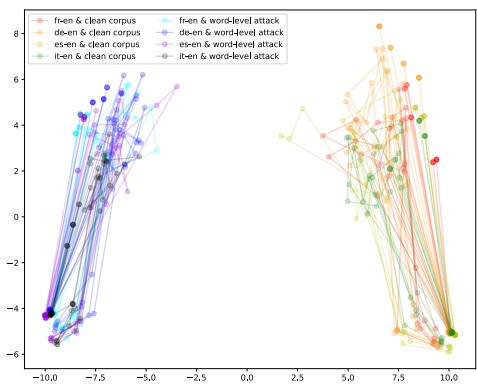

Figure 4: PCA-based dimension-reduced representations of the encoder outputs of sentences with character-level noise. The encoders are from a model trained on a clean corpus and a model trained on character-level attacks respectively.

Figure 5: PCA-based dimension-reduced representations of the encoder outputs of sentences with word-level noise. The encoders are from a model trained on a clean corpus and a model trained on word-level attacks respectively.

bations in the input. Moreover, the models trained with character-level attacks are more aligned in terms of noisy semantic representations across different translation directions. This alignment could be a crucial factor contributing to the transfer of robustness across translation directions. Throughout the model training process, the model needs to learn the correspondence between the source language and the target language. The noise generated by the black-box attacks compels the model to capture alignment information of semantic units to better handle the noise. This semantic alignment assists the model in effectively dealing with noisy data in other translation directions.

Similarly, we employed word-level black-box attack methods to generate four word-level noisy sentences for each source language's seed sentence. Each of these four sentences corresponds to four word-level attack operations, including random word swapping, random word deletion, insertion and replacement operations using word embeddings with words similar to the target word. We fed them into models trained on the clean corpus, as well as models trained on word-level attacks for specific translation directions. Figure 5 displays dimension-reduced representations of the encoder outputs of these two models using PCA. The same conclusion can be observed for the representations of word-level noisy sentences as for those of character-level noisy sentences. This further substantiates the presence of robustness transfer in multilingual NMT. The continuous alignment

of the noise semantic representations throughout training facilitates the transfer of model robustness towards noise across translation directions.

At the end we conducted additional experiments to investigate the effect of shared encoder on robustness transfer. The results and conclusions are presented in the Appendix A.

### 4.3 Robustness to Character-Level Noise Tends to Transfer across Related Languages

We further investigated the robustness transfer related to character-level noise across languages. As depicted in Table 2, it is evident that the model trained with character-level attacks on the French-English translation direction exhibits a greater enhancement in BLEU scores on the character-level noise test set compared to the model trained with word-level attacks tested on the word-level noise test set. This finding suggests a higher likelihood of robustness transfer for character-level noise across languages within this experimental setup.

Let's revisit the results in the experimental settings involving many-to-one scenarios where the source languages are from different language families, as depicted in Table 3. We observe that the previously proposed conclusion does not hold in these cases. We hypothesize that the conclusion is applicable only to related languages, e.g., languages from the same language family. We can also validate the applicability of this conclusion using the results obtained from our one-to-many

| | DE-EN |
|---|---|
| sentence without noise | Wir hatten Angst, aber wir wollten trotzdem zur Schule gehen. |
| reference translation | We were scared, but still, school was where we wanted to be. |
| sentence with character-level noise | Wir thatten Angst, aber wir woJllten trotzdem zur Schule gehen. |
| hypothesis translation (model trained with character-level attacks) | We fear that, but we still wanted to go to school. |
| sentence with word-level noise | Wir hatten Angst, aber gehörte wollten trotzdem wirkungsgeschichte Schule gehen. |
| hypothesis translation (model trained with word-level attacks) | We were scared, but we wanted to go to real school anyway. |

Table 4: Translation examples of related languages (from German to English).

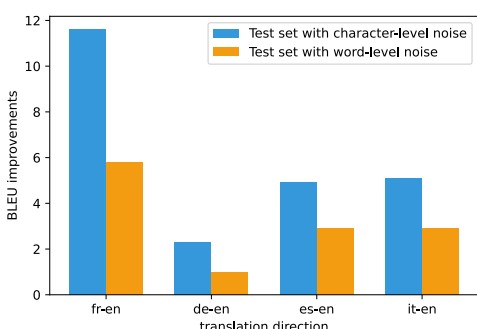

Figure 6: BLEU improvements on the test set with character-/word-level noise for models trained on character-/word-level attacks on the French-English translation direction over models trained on the clean corpus. The dataset used is News Commentary. The source languages are from the same language family.

experiments, as presented in Table 1. When the source language is English for all translation directions, which increases the similarity in translation directions, the experimental outcomes also indicate a greater likelihood of robustness transfer for character-level noise across languages.

The aforementioned experimental conclusions are drawn from experiments conducted on the TED TALKS dataset. In order to ensure the reliability of the findings, we also conducted experiments on the News Commentary dataset. These experiments were carried out in a many-to-one setting where all source languages are from the same language family. During training, we attacked the French-English translation direction. Results are shown in Figure 6. It can be observed that the model trained with character-level attacks exhibits a similarly larger improvement in BLEU on the character-level noise test set. This further emphasizes that the robustness of the model towards character-level noise is more likely to transfer across related languages.

To provide a more intuitive illustration of this finding, a case study has been conducted. Table 4 presents translation examples from German to English. It is evident that when the source and target languages are closely related, inputting sentences with character-level noise into the model trained with character-level attacks yields superior translation quality compared to inputting sentences with word-level noise into the model trained with word-level attacks.

We speculate that the reason for this may be due to the close sharing of commonality in vocabulary, grammar structures, and language features among related languages. The model can capture character-level similarities between different languages, making it easy for the model's robustness against character-level noise to transfer between related languages.

### 4.4 Robustness to Word-Level Noise Tends to Transfer across Distant Languages

On the other hand, we also further investigated the transfer of robustness towards word-level noise across languages. In Table 3, it can be observed that the models trained with word-level attacks on the French-English translation direction gain a greater improvement in BLEU on the word-level noise test set in the experimental setup where the source languages are from different language families. Additionally, we also note that models trained with word-level attacks outperform models trained with character-level attacks on the not-being-attacked translation directions of the multi-level noise test set. This suggests that, in this experimental setup, the robustness of the model towards word-level noise is more likely to transfer to other translation directions than towards character-level noise.

We have observed a similar experimental phenomenon in cases where the source languages are in the same language family. When the attacked

| | **ZH-EN** |
|---|---|
| sentence without noise
reference translation | 为了 解释 这个 ， 我 要说 一件 很 让 人 困扰 的 事实
Now , to explain this , I need to tell you a very disturbing fact. |
| sentence with character-level noise
hypothesis translation
(model trained with character-level attacks) | 为了 解释 这个 ， 我 要说 t件 很 让 人 困扰 的 事

To explain this, I'm going to talk about everything that's disturbing. |
| sentence with word-level noise
hypothesis translation
(model trained with word-level attacks) | 为了 解释 这个 ， 要说 一件 让 很 人 困扰 事实

To explain this, I'm going to tell you one thing that's very troubling. |

Table 5: Translation examples of distant languages (from Chinese to English).

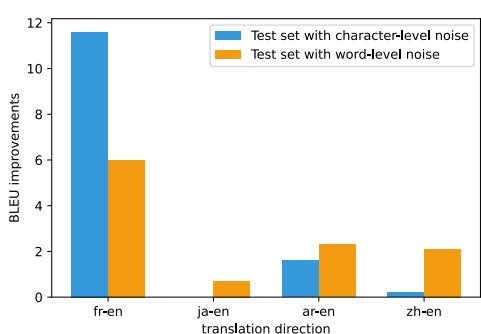

Figure 7: BLEU improvements on the test set with character-/word-level noise for models trained on character-/word-level attacks on the French-English translation direction over models trained on the clean corpus. The dataset used is News Commentary and KFTT. The source languages are from different language families.

translation direction is German-English, the robustness of the model towards word-level noise is more likely to transfer across translation directions. This observation leads us to speculate that the robustness of the model to word-level noise is more likely to transfer across distant languages, which is supported by the fact that German is in a different language branch from the source languages of the other three translation directions.

To further validate this, we also conducted experiments on the News Commentary dataset. In particular, the KFTT dataset was chosen as the training data for the Japanese-English translation direction to compensate for the lack of Japanese-English parallel data in the News Commentary dataset. Results are shown in Figure 7. We observe that, apart from the attacked translation direction, the BLEU improvement on the word-level noise test set is greater than that on the character-level noise test set for all other translation directions. This provides stronger

evidence that the robustness of the model towards word-level noise is more likely to transfer across dissimilar translation directions. It is worth noting that the transfer degree may be slightly weaker in Japanese-English translation direction as it is a different domain of data from those of other translation directions. Nevertheless, robustness is still able to transfer across languages despite domain shift.

We also provide examples of zh-en translation direction to demonstrate this observation in Table 5. It can be seen that the model trained on word-level noise attack behaves better when dealing with word-level noise.

We assume that the reason for this is that in the case of distant languages, compared to character-level alignment, the model may be more inclined to perform semantic alignment at the word level. Therefore, robustness to word-level noise is more likely to transfer across languages.

## 5 Conclusions

In this paper, we have conducted an extensive and in-depth investigation on robustness transfer in multilingual neural machine translation models with the proposed robustness transfer analysis protocol. Our empirical findings demonstrate that robustness can indeed transfer across languages. Furthermore, we find that robustness towards character-level noise is more likely to transfer across related language while robustness towards word-level noise is more likely transfer across distant languages.

## Limitations

Findings in this paper are derived from empirical experiments while underlying reasons for these findings are yet to be further investigated. In addition, noises in this study are synthetic. Naturally occurring noise encompasses a broader range of

variations, necessitating further investigation into the transfer patterns of robustness. Finally, other numerous factors potentially influence the transfer of robustness across translation directions. This paper focuses on a subset of such factors, including language distance and noise types, leaving room for the exploration of additional influential factors in the future.

## Ethics Statement

This study adheres to the ethical guidelines set forth by our institution and follows the principles outlined in the ACM Code of Ethics and Professional Conduct. All datasets used in our experiments are publicly available.

## Acknowledgments

The present research was supported by the Natural Science Foundation of Xinjiang Uygur Autonomous Region (No. 2022D01D43) and the Key Research and Development Program of Yunnan Province (No. 202203AA080004). We would like to thank the anonymous reviewers for their insightful comments.

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

| Training dataset | Test dataset | fr-en | de-en | es-en | it-en |
|---|---|---|---|---|---|
| clean corpus | clean corpus | **41.6** | 36.4 | 42.1 | **38.8** |
| | character-level attack | 23.7 | 23.3 | 26.6 | **25.2** |
| | word-level attack | 27.2 | 24.7 | 28.5 | **26.6** |
| | multi-level attack | 24.8 | 23.8 | 27.2 | 25.2 |
| character-level attack | clean corpus | 41.1 | **36.8** | **42.3** | **38.8** |
| | character-level attack | **37.7** | 23.0(↓1.3%) | 26.8(↑0.7%) | 24.9(↓1.2%) |
| | word-level attack | 29.3 | 24.1 | 29.1 | 26.4 |
| | multi-level attack | 32.8 | 23.3 | 27.4 | **25.3** |
| word-level attack | clean corpus | 40.8 | 36.8 | 42.2 | 38.8 |
| | character-level attack | 28.1 | **23.7** | **27.1** | 24.8 |
| | word-level attack | **35.3** | 24.8(↑0.4%) | 29.4(↑3.2%) | 26.3(↓1.1%) |
| | multi-level attack | 31.2 | 23.7 | **27.7** | 25.0 |
| multi-level attack | clean corpus | 40.5 | 36.7 | 42.2 | 38.5 |
| | character-level attack | 35.8 | 23.6 | 27.0 | 24.6 |
| | word-level attack | 34.4 | **24.8** | 29.2 | 26.2 |
| | multi-level attack | **34.7** | 24.0(↑0.8%) | 27.7(↑1.8%) | 25.1(↓0.4%) |

Table 6: BLEU scores for models with different training settings on different test sets using the TED TALKS dataset in the many-to-one case. The bolded scores represent the optimal performance on each test set for a specific language direction. The source languages are from the same language family. The attacked translation direction is fr-en. Encoders are not shared across source languages.

## A  Additional Experiments: No Shared Encoder across Source Languages

We conducted additional experiments to demonstrate the significance of the shared encoder in enabling robustness transfer. The experimental results are shown in Table 4. It can be observed that when source languages do not share the same encoder, robustness barely transfers across translation directions.

## B  Model Hyperparameters

| Model Hyperparameters | Value |
|---|---|
| adam-betas | (0.9, 0.98) |
| lr | 0.0005 |
| warmup-updates | 4000 |
| warmup-init-lr | 1e-07 |
| label-smoothing | 0.1 |
| dropout | 0.3 |
| weight-decay | 0.0001 |
| max-tokens | 4000 |
| update-freq | 8 |

Table 7: Model Hyperparameters.