# OpenReview forum: "Is Robustness Transferable across Languages in Multilingual Neural Machine Translation?"
_EMNLP/2023/Conference — EMNLP 2023 Findings_

### Official Review · Reviewer_SHa5 · 2023-07-31

**Soundness:** 3

**Excitement:**

2: Mediocre: This paper makes marginal contributions (vs non-contemporaneous work), so I would rather not see it in the conference.

**Missing References:**

* Line 094 (full parameter-sharing): Johnson et al. Google’s multilingual neural machine translation system: enabling zero-shot translation. In: TACL 2017

**Paper Topic And Main Contributions:**

This analysis paper investigates robustness in the context of multilingual machine translation. In particular, the authors investigate the effect of different types of noise (insertion of character-level noise / word-level noise / the combination thereof) on model performance.

The results seem to indicate that (unsurprisingly) robustness transfers, where the magnitude of transfer is relatively high for one-to-many models and relatively low for many-to-one models.

**Questions For The Authors:**

A. Do you have a hypothesis that explains why the degree of robustness transfer does not seem to rely on source language similarity?

B. How do you combine the pretrained GloVe / fasttext word embeddings that are used for word-level attacks with your model, which has a different vocabulary than GloVe? Do you only use the subset which matches? If so, what percentage of the total vocab is this?

**Reasons To Accept:**

* The investigation of robustness transferability has not been explored before.
* The results clearly indicate that robustness is transferable. This finding can be used by follow-up work that aims to improve MNMT.

**Reasons To Reject:**

* The connection to related work is insufficient.

Apart from the observation that the authors investigate robustness in a multilingual setting, and previous work considered a bilingual setting, it does not become clear how this work fits in with the broader field. Black-box methods and white-box methods are mentioned, but not explained. No attempt is made to put the experimental results in context with earlier findings of the field. The paper of the quality would improve with some more discussion: which findings are in agreement with earlier work? Which disagree, and what are the possible reasons? From the paper, it's also not clear how the types of noise that you consider are (dis)similar to earlier work.

* The findings lack depth

Most of the findings are unsurprising, which is of course not necessarily an issue. Unfortunately, for the surprising findings, there is no further investigation. For instance, comparing Table 2 and Table 3, the authors find that the degree of robustness transfer does not seem to rely on source language similarity, which is a highly surprising finding. I think the paper would benefit a lot from digging into this deeper.

* Important information is missing. Presentation could use improvement.

It is not clear what data is used exactly. Authors mention TED Talks and News Commentary, but it is not clear which experiments use which dataset. Maybe they are combined? It is also not clear which test set is used. TED Talks, News Commentary, or the combination thereof?

It is unclear how to interpret "bold" in the Table; this is not mentioned anywhere and it's not as simple as "best score".

The differences in Figure 4 and Figure 5 are very subtle, and based on these plots I would not conclude that representations learned using noisy data is more compact compared to representations trained on clean data. It would be easy to quantify this claim by calculating representational differences, which could be used as additional evidence.

The BLEU signature is missing.

**Reproducibility:**

2: Would be hard pressed to reproduce the results. The contribution depends on data that are simply not available outside the author's institution or consortium; not enough details are provided.

**Reviewer Confidence:**

4: Quite sure. I tried to check the important points carefully. It's unlikely, though conceivable, that I missed something that should affect my ratings.

**Typos Grammar Style And Presentation Improvements:**

* Line 078: "For summary" -> "To summarize"
* Figures 2 and 3 are hard to read, and I would argue that this type of Figure is not well-suited for the message you're trying to convey. A Table would be a better fit. If you insist on a Figure, I would go for a radar chart.
* Line 214: it would be a nice courtesy to the reader to summarise model hyperparameters here (or in Appendix).

---

> ### Author Rebuttal · Authors · 2023-08-29
>
> Many thanks for your insightful comments and suggestions. We sincerely appreciate your time in reviewing the paper, and our point-to-point responses to your comments are given below.
>
> > Apart from the observation that the authors investigate robustness in a multilingual setting, and previous work considered a bilingual setting, it does not become clear how this work fits in with the broader field. Black-box methods and white-box methods are mentioned, but not explained. No attempt is made to put the experimental results in context with earlier findings of the field. The paper of the quality would improve with some more discussion: which findings are in agreement with earlier work? Which disagree, and what are the possible reasons? From the paper, it's also not clear how the types of noise that you consider are (dis)similar to earlier work.
>
> Thank you for the advice. Previous studies have shown that black-box attacks can indeed improve the robustness of machine translation models[1], which is consistent with our observations. In addition, the attacks used in this paper include three types, including character-level[2], word-level, and multi-level attacks. The character-level attacks include random character insertion, deletion, replacement, and adjacent character swap. The word-level attacks include random word swapping, deletion[3], and similar word insertion and replacement[4]. The multi-level attacks combine the first two attack methods. We will include the above in more details in the Related Work section of our paper. For comparison to previous findings, it might be difficult as we conduct experiments in the context of multilingual machine translation and try to find robustness patterns across multiple translation directions while previous works focus on bilingual machine translation where there is only one translation direction. This makes a direct comparison to previous findings lack of foundation.
>
> [1]Belinkov Y, Bisk Y. Synthetic and Natural Noise Both Break Neural Machine Translation[C]//International Conference on Learning Representations. 2018.
>
> [2]Karpukhin V, Levy O, Eisenstein J, et al. Training on Synthetic Noise Improves Robustness to Natural Noise in Machine Translation[C]//Proceedings of the 5th Workshop on Noisy User-generated Text (W-NUT 2019). 2019: 42-47.
>
> [3]Wei J, Zou K. EDA: Easy Data Augmentation Techniques for Boosting Performance on Text Classification Tasks[C]//Proceedings of the 2019 Conference on Empirical Methods in Natural Language Processing and the 9th International Joint Conference on Natural Language Processing (EMNLP-IJCNLP). 2019: 6382-6388.
>
> [4]Alzantot M, Sharma Y, Elgohary A, et al. Generating Natural Language Adversarial Examples[C]//Proceedings of the 2018 Conference on Empirical Methods in Natural Language Processing. 2018: 2890-2896.
>
> > Most of the findings are unsurprising, which is of course not necessarily an issue. Unfortunately, for the surprising findings, there is no further investigation. For instance, comparing Table 2 and Table 3, the authors find that the degree of robustness transfer does not seem to rely on source language similarity, which is a highly surprising finding. I think the paper would benefit a lot from digging into this deeper.
>
> Thanks for this comment. We would like to point out, however, that the statement that robustness transfer does not depend on the similarity of the source language is not actually the view expressed in this paper. It will be specified in a later answer to your question.
>
> > It is not clear what data is used exactly. Authors mention TED Talks and News Commentary, but it is not clear which experiments use which dataset. Maybe they are combined? It is also not clear which test set is used. TED Talks, News Commentary, or the combination thereof?
>
> For 4.2 experiments, we used the TED Talks dataset for training. In 4.3 experiments, we used News Commentary dataset. For 4.4 experiments, we used News Commentary dataset for fr-en, ar-en, and zh-en translation directions and KFTT dataset for ja-en translation. We will explain more clearly about datasets used in the Experiment Settings part of our revised paper.
>
> > It is unclear how to interpret "bold" in the Table; this is not mentioned anywhere and it's not as simple as "best score".
>
> In fact, the bolded scores represent the optimal performance on each test dataset within a specific language direction. For instance, as shown in Table 2 for the fr-en translation, the highest BLEU score for the clean corpus dataset is attained when the model is trained with the corresponding clean corpus, resulting in the bolded value of 41.9. Further elucidation regarding the significance of these bolded scores in the table will be meticulously provided in our revised paper.
>
> > The differences in Figure 4 and Figure 5 are very subtle, and based on these plots I would not conclude that representations learned using noisy data is more compact compared to representations trained on clean data. It would be easy to quantify this claim by calculating representational differences, which could be used as additional evidence.
>
> Thank you for your feedback. We have chosen models trained with character-level noise attacks and models trained on clean corpus. We input sentences containing character-level noise into the models and measure the representation differences in the encoder outputs by calculating the Euclidean distance between the output vectors of the model's encoder for each pair of languages. The average representation difference between each pair of languages is then computed. The same procedure is applied for word-level noise to calculate the Euclidean distance between the noise representations. After conducting these calculations, the resulting data is presented in the table below.
>
> |                             | models trained with character-level noise attacks | models trained with word-level noise attacks | models trained with clean data |
> | --------------------------- | ------------------------------------------------- | -------------------------------------------- | ------------------------------ |
> | character-level noise input | 109.4941                                          | -                                            | 113.6249                       |
> | word-level noise input      | -                                                 | 113.3749                                     | 115.7528                       |
>
> From the table, we can conclude that when we input noise sentences from each language into the corresponding noise attack trained models, the noise representations produced by different languages are indeed more compact.
>
> > The BLEU signature is missing.
>
> We use sacrebleu to calculate the BLEU score. The details will be added to the revised paper.
>
> > Do you have a hypothesis that explains why the degree of robustness transfer does not seem to rely on source language similarity?
>
> Thank you for your question. However, we would like to clarify a potential misunderstanding of the point our work aims to convey. Our findings do not suggest that robustness transfer is independent of the similarity between source languages. On the contrary, a comparison between Figure 6 and Figure 7 in the paper reveals that both character-level and word-level noise robustness exhibit greater improvements under experimental settings where the source languages belong to the same language family.
>
> Our intended message is that, among similar languages, the robustness of a model to character-level noise is more likely to be transferred between translation directions than the robustness to word-level noise. Conversely, across distant language pairs, the model's robustness to word-level noise is more likely to be transferred between translation directions than the robustness to character-level noise. This phenomenon could be attributed to the degree of alignment between languages. Similar languages might share a greater portion of vocabulary, grammatical structures, and linguistic traits, enabling the model to capture similarities at the character level across different languages. Hence, the model's robustness to character-level noise might be more readily transferrable between languages. In the case of distant languages with greater dissimilarity, the model might rely more on word-level alignment to fulfill translation tasks. Consequently, the robustness to word-level noise is more prone to transfer across distant languages.
>
> > How do you combine the pretrained GloVe / fasttext word embeddings that are used for word-level attacks with your model, which has a different vocabulary than GloVe? Do you only use the subset which matches? If so, what percentage of the total vocab is this?
>
> Thank you for this question. Word embeddings are employed to identify semantically related words for word-level attack (replacement or insertion). GloVe word embeddings are utilized for English texts, whereas fastText word embeddings specific to each language are employed for texts in other languages. It's worth noting that these embeddings are not combined. The aforementioned word replacement and insertion attack method is implemented based on the nlpaug library. We first find the top-k words with similar word embeddings as the target word and then randomly select 1 word from them as the word for replacement or insertion.
>
> > Line 078: "For summary" -> "To summarize"
> >
> > Figures 2 and 3 are hard to read, and I would argue that this type of Figure is not well-suited for the message you're trying to convey. A Table would be a better fit. If you insist on a Figure, I would go for a radar chart.
> >
> > Line 214: it would be a nice courtesy to the reader to summarise model hyperparameters here (or in Appendix).
>
> Thank you for pointing out these issues. We also agree that radar chart is better to represent the data rather than line plot. We will consider changing the figure into radar chart. Regarding model hyperparameters, we basically used the default parameters of the multilingual_transformer_iwslt_de_en model provided by the fairseq framework. The parameters that need to be additionally pointed out to be set in the training script are shown in the following table. These hyperparameter settings will be added to the appendix of the next version.
>
> | hyperparameter      | value |
> | --------------- | ----------- |
> | adam-betas      | (0.9, 0.98) |
> | lr              | 0.0005      |
> | warmup-updates  | 4000        |
> | warmup-init-lr  | 1e-07       |
> | label-smoothing | 0.1         |
> | dropout         | 0.3         |
> | weight-decay    | 0.0001      |
> | max-tokens      | 4000        |
> | update-freq     | 8           |

---

### Official Review · Reviewer_USAS · 2023-08-01

**Typos Grammar Style And Presentation Improvements:** 1. Line 075, interested -> interested…
**Soundness:** 3

**Excitement:**

3: Ambivalent: It has merits (e.g., it reports state-of-the-art results, the idea is nice), but there are key weaknesses (e.g., it describes incremental work), and it can significantly benefit from another round of revision. However, I won't object to accepting it if my co-reviewers champion it.

**Paper Topic And Main Contributions:**

This paper aims to study whether robustness can be transferred across different languages in multilingual neural machine translation (MNMT) and the answer may be yes. The authors design several attack experiments for one-to-many and many-to-one MNMT models and obtain consistent conclusions. These findings are somewhat helpful for understanding the robustness transfer in MNMT models and multilingual encoder models.

**Questions For The Authors:**

- Line 288, what does "In order to avoid experimental chance," mean? Please change a more precise expression.

**Reasons To Accept:**

- The authors conduct a series of analyses on the core research question, from shallow to deep, providing sufficient (some even redundant) evidence for their claim.
- The findings may motivate more future work on the robustness of the multilingual encoders (known to be vulnerable as QE models).

**Reasons To Reject:**

- Many figures in the paper lead to similar conclusions (e.g., Figure 2 vs. Figure 3, Figure 4 vs. Figure 5), which take up lots of space in the paper and make the paper look verbose. It would be better to move some of them to the appendix.
- Figure 4,5 look a little confusing and need more detailed descriptions.
- For most translation directions, the robustness transferred from EN-FR is not significant enough (still lags behind the results on clean corpus by ≈10 BLEU scores). Although the authors use growth rates to make the benefits look more obvious in Tables 1,2,3, the transfer effects are still limited.
- I would like to see more results based on multilingual pre-trained models, e.g., mBART (Liu et al., 2020), mRASP2 (Pan et al., 2021).



1. Liu et al. Multilingual Denoising Pre-training for Neural Machine Translation. TACL 2020.
2. Pan et al. Contrastive Learning for Many-to-many Multilingual Neural Machine Translation. ACL 2021.

**Reproducibility:**

4: Could mostly reproduce the results, but there may be some variation because of sample variance or minor variations in their interpretation of the protocol or method.

**Reviewer Confidence:**

4: Quite sure. I tried to check the important points carefully. It's unlikely, though conceivable, that I missed something that should affect my ratings.

---

> ### Author Rebuttal · Authors · 2023-08-29
>
> We greatly appreciate your valuable insights and suggestions. We have taken meticulous care to integrate these insights into the revised version of our paper. For ease of your reference, we present your comments first, followed by our detailed responses addressing each point raised.
>
> > Many figures in the paper lead to similar conclusions (e.g., Figure 2 vs. Figure 3, Figure 4 vs. Figure 5), which take up lots of space in the paper and make the paper look verbose. It would be better to move some of them to the appendix.
>
> Thank you for this feedback. Figure 3 and Figure 5 visualize results when attacking the MNMT model with word-level attack. As the results are relatively similar to the character-level attack, we will consider moving Figure 3 and Figure 5 to the appendix following your suggestion.
>
> > Figure 4,5 look a little confusing and need more detailed descriptions.
>
> Figures 4 and 5 visualize the noise representations of a model trained on a clean corpus and a model trained on noise attack. The primary objective is to assess the alignment of noise representations across different languages. Taking Figure 4 as an example, we started by selecting semantically equivalent seed sentences for each source language to maximize the similarity of their semantic representations. Subsequently, based on these seed sentences, we employed the four character-level black-box attack methods in our paper to generate four distinct character-level noisy sentences, each corresponding to a specific attack method. Next, we combined noisy sentences with clean sentences and fed them into two models: one trained on the clean corpus and the other trained on character-level attacks on the French-English translation direction. Subsequently, we applied PCA dimensionality reduction to the outputs of the encoders of both models and visualize the reduced representations of the noisy input. Figure 5 essentially involves a similar procedure. The difference lies in the fact that the four noisy sentences correspond to four types of word-level attack methods, using models trained on clean corpus and models trained with word-level attacks in the specific direction. We'll make these more clear in the next version with detailed descriptions as you suggested.
>
> > For most translation directions, the robustness transferred from EN-FR is not significant enough (still lags behind the results on clean corpus by ≈10 BLEU scores). Although the authors use growth rates to make the benefits look more obvious in Tables 1,2,3, the transfer effects are still limited.
>
> Thank you for this question. The approach employed in this paper merely illustrates the transferability of robustness across different translation directions, leaving substantial room for further improvements in the effectiveness of robustness transfer that can be achieved. In our upcoming research, we will delve deeper into enhancing the transfer of robustness across various translation directions.
>
> > I would like to see more results based on multilingual pre-trained models, e.g., mBART (Liu et al., 2020), mRASP2 (Pan et al., 2021).
>
> Our findings have implications for machine translation with either solely multilingual LLMs or models enhanced by multilingual LLMs. We'd like to extend our framework to investigate the robustness transfer across languages with multilingual LLMs in our future work.
>
> > Line 288, what does "In order to avoid experimental chance," mean? Please change a more precise expression.
>
> We would like to explain about additional experiments for supporting our findings. We will change this sentence into "For improving the reliability of our findings".

---

### Official Review · Reviewer_xN8k · 2023-08-02

**Soundness:** 4

**Excitement:**

3: Ambivalent: It has merits (e.g., it reports state-of-the-art results, the idea is nice), but there are key weaknesses (e.g., it describes incremental work), and it can significantly benefit from another round of revision. However, I won't object to accepting it if my co-reviewers champion it.

**Paper Topic And Main Contributions:**

This paper systematically investigates whether the robustness of multilingual machine translation (MNMT) models can be transferred between different languages, via crafting different adversarial examples towards various language pairs to attack MNMT models. Experimental results demonstrate that robustness of MNMT models can transfer across different languages.

**Questions For The Authors:**

A. What's the potential application of this paper? Could it provide insights regarding how to enhance the translation robustness of LLMs?

B. Could you provide some cases in section 4.3 and 4.4?

**Reasons To Accept:**

1. The motivation of this paper is clear.
2. The proposed experiments illustrate some insightful findings.

**Reasons To Reject:**

1. Although adversarial attacks enhance the robustness of MNMT models, the sacrifice on clean test set seems not trivial (e.g., 1~2 BLEU score on the en-fr translation task in Table1). It may be helpful to also investigate the trade-off between robustness and clean-performances of MNMT models.

2. Some cases would be helpful when explaining the findings in section 4.3 and 4.4, i.e., why character-level attack is more useful for enhancing MNMT model robustness in related languages?

**Reproducibility:**

4: Could mostly reproduce the results, but there may be some variation because of sample variance or minor variations in their interpretation of the protocol or method.

**Reviewer Confidence:**

4: Quite sure. I tried to check the important points carefully. It's unlikely, though conceivable, that I missed something that should affect my ratings.

---

> ### Author Rebuttal · Authors · 2023-08-29
>
> Thank you so much for your insightful comments and suggestions, and they are exceedingly helpful for us to improve our paper. We will earnestly incorporate them into the revised version of our paper. In the following, your comments are first stated and then followed by our point-by-point responses.
>
> > Although adversarial attacks enhance the robustness of MNMT models, the sacrifice on clean test set seems not trivial (e.g., 1~2 BLEU score on the en-fr translation task in Table1). It may be helpful to also investigate the trade-off between robustness and clean-performances of MNMT models.
>
> Thank you for this constructive comment. We do notice that when attacking a certain translation direction, the performance of that direction on the clean corpus decreases significantly, and we will continue our research to improve the robustness of the model while maintaining its performance on the clean corpus. However, we are also delighted to note that the noise attack on one translation direction does not seem to significantly affect the performance of other translation directions on the clean corpus, and even the performance of other translation directions on both the clean corpus and the noise test set is improved at the same time.
>
> > What's the potential application of this paper? Could it provide insights regarding how to enhance the translation robustness of LLMs?
>
> The main motivation of our work is to try to find the robustness patterns and mechanisms of multilingual machine translation, which finally helps the development of efficient approaches to improving translation robustness across multiple languages in multilingual machine translation. We find that when training a multilingual neural machine translation (MNMT) model using a black-box attack method for a single translation direction, translation robustness is also improved for all or related translation directions. With this finding, it is hoped that it will (1) improve the robustness of translations with optimized attacking efforts and (2) improve translation robustness of languages (e.g., low-resource languages) where noisy datasets are not easy to collect with noisy data from languages where those datasets are available. We believe that our work has implications for the translation robustness of multilingual LLMs as they also share the same model for all languages. For example, where we might be able to improve robustness across all languages by adding high-quality noise data only to high-resource languages. We'd like to provide such new results on LLMs in the next version following your great suggestion.
>
> > Could you provide some cases in section 4.3 and 4.4?
>
> In order to illustrate the issue more intuitively, a case study has been conducted following your suggestion. First, some examples of de-en translation directions are provided in the following Table 1 to demonstrate the conclusion in Section 4.3 that the robustness of the model to character-level noise transfers more easily between translation directions than robustness to word-level noise among related languages. It can be observed from the data in Table 1 that the models trained on character-level noise attacks perform better when dealing with character-level noise.
>
> |                                                              | DE-EN                                                        |
> | ------------------------------------------------------------ | ------------------------------------------------------------ |
> | sentence without noise                                       | Wir hatten Angst, aber wir wollten trotzdem zur Schule gehen. |
> | reference translation                                        | We were scared, but still, school was where we wanted to be. |
> | sentence with character-level noise                          | Wir thatten Angst, aber wir woJllten trotzdem zur Schule gehen. |
> | translation results of feeding sentences with character-level noise into the model trained with the character-level noise attack | We fear that, but we still wanted to go to school.           |
> | sentence with word-level noise                               | Wir hatten Angst, aber gehörte wollten trotzdem wirkungsgeschichte Schule gehen. |
> | translation results of feeding sentences with word-level noise into the model trained with the word-level noise attack | We were scared, but we wanted to go to real school anyway .  |
>
> Table 1. Examples of translations with source languages being related languages
>
> In addition, we also provide examples of zh-en translation direction to demonstrate the conclusions in Section 4.4. Between languages that are distant from each other, the robustness of the model to word-level noise is more likely to transfer between translation directions than the robustness to character-level noise. It can be observed from the data in Table 2 that in this case the model trained on word-level noise attack behaves better when dealing with word-level noise.
>
> |                                                              | ZH-EN                                                        |
> | ------------------------------------------------------------ | ------------------------------------------------------------ |
> | sentence without noise                                       | 为了 解释 这个 ，我 要说 一件 很 让 人 困扰 的 事实          |
> | reference translation                                        | Now , to explain this , I need to tell you a very disturbing fact . |
> | sentence with character-level noise                          | 为了 解释 这个 ， 我 要说 t件 很 让 人 困扰 的 事            |
> | translation results of feeding sentences with character-level noise into the model trained with the character-level noise attack | To explain this, I'm going to talk about everything that's disturbing . |
> | sentence with word-level noise                               | 为了 解释 这个 ， 要说 一件 让 很 人 困扰 事实               |
> | translation results of feeding sentences with word-level noise into the model trained with the word-level noise attack | To explain this, I'm going to tell you one thing that's very troubling. |
>
> Table 2. Examples of translations with source languages being distant languages

---

### Meta-Review · Area_Chair_MHLS · 2023-09-18

**Recommendation:** 3

**Metareview:**

This paper studies whether robustness to input noise, as conferred by “attacks” involving noised input during training, can be transferred across languages in multilingual NMT. Experimental results indicate that this can indeed be the case, and further analyses characterize the conditions under which it occurs.

Reviewers felt that the study was novel and well motivated, and were mostly convinced by the findings and analyses indicating positive results. They noted some downsides, including a tradeoff between robustness and quality (which is however confined to the language under attack), some problems with the clarity and conciseness of the presentation, and a lack of further analysis to account for some of the experimental results.

This is a solid contribution. As the authors point out, a big benefit to their findings is that robustness to a high-resource language can be transferred to a lower-resource language without needing to explicitly attack the data in that language, which might be problematic for various reasons (including the potential to reduce output quality that might already be marginal). The paper has the potential to generate interesting follow-up work, in particular to try to solve the robustness / quality tradeoff for languages subjected to attacks.

---

### Decision · Program_Chairs · 2023-10-07

**Decision:**

Accept-Findings

**Comment:**

This paper studies whether robustness to input noise, as conferred by “attacks” involving noised input during training, can be transferred across languages in multilingual NMT. Experimental results indicate that this can indeed be the case, and further analyses characterize the conditions under which it occurs.

Reviewers felt that the study was novel and well motivated, and were mostly convinced by the findings and analyses indicating positive results. They noted some downsides, including a tradeoff between robustness and quality (which is however confined to the language under attack), some problems with the clarity and conciseness of the presentation, and a lack of further analysis to account for some of the experimental results.

This is a solid contribution. As the authors point out, a big benefit to their findings is that robustness to a high-resource language can be transferred to a lower-resource language without needing to explicitly attack the data in that language, which might be problematic for various reasons (including the potential to reduce output quality that might already be marginal). The paper has the potential to generate interesting follow-up work, in particular to try to solve the robustness / quality tradeoff for languages subjected to attacks.